# Self-Regulation of Soil Enzyme Activity and Stoichiometry under Nitrogen Addition and Plastic Film Mulching in the Loess Plateau Area, Northwest China

Meixia Liu [1], Menglu Wang [1,2], Congwei Sun [1,2], Hui Wu [1], Xueqing Zhao [1], Enke Liu [1,2], Wenyi Dong [1,*] and Meiling Yan [3,*]

1 Institute of Environment and Sustainable Development in Agriculture, Chinese Academy of Agricultural Sciences, Beijing 100081, China
2 Key Laboratory of Dryland Agriculture, Ministry of Agriculture and Rural Affairs of the People's Republic of China, Beijing 100081, China
3 Yantai Academy of Agricultural Sciences, Yantai 265500, China
* Correspondence: dongwenyi@caas.cn (W.D.); mlyan-2004@163.com (M.Y.)

**Abstract:** Soil extracellular enzyme activity (EA) and its eco-enzyme stoichiometric ratio (ES) are extremely sensitive to environmental change. This study aimed to clarify the change law of EA and ES in soil with different nitrogen addition levels under plastic film mulching, and to optimize the application amount of nitrogen fertilizer that was used. Based on the location experiment of plastic film mulching fertilization that has been ongoing since 2015, soil samples were collected from different depths (0–10 cm, 10–20 cm and 20–30 cm) during the harvest period of spring maize in October 2021. Four soil extracellular enzyme activities (β-1,4 glucosidase (βG), β-1, 4-N-acetylglucosidase (NAG), leucine aminopeptidase (LAP) and alkaline phosphatase (AP)) involved in soil carbon (C), nitrogen (N) and phosphorus (P) cycling at different nitrogen application levels (0, 90, 150, 225 and 300 kg·hm$^{-2}$) were studied under two planting patterns of no plastic film mulching (LD) and plastic film mulching (PM). The latest discovery of this study is that the activities of soil EA involved in the cycling of soil carbon C, N and P are similar in different soil depths (0–10 cm, 10–20 cm and 20–30 cm). Both $E_{C:P}$ and $E_{C:P}$ in the soil in this area are less than 1:1, indicating that the soil is limited by N and P. Comprehensive analysis showed that a nitrogen application level of 225 kg·hm$^{-2}$ was beneficial to the balance of soil nutrients and the improvement of soil EA at harvest. At the same time, PM can effectively improve the soil EA and is more conducive to the balance of soil nutrients. Redundancy analysis (RDA) showed that EA and ES were strongly correlated with pH, soil organic carbon (SOC), total nitrogen (TN) and total phosphorus (TP). Most importantly, this study revealed that the activity of extracellular enzymes in arid and semi-arid areas was constantly self-regulated with the addition of nitrogen, which provided theoretical and technical support for the efficient use of nitrogen under the condition of plastic film mulching.

**Keywords:** plastic film mulching; N addition levels; different soil depths; soil extracellular enzyme activity; ecological stoichiometric ratio




## 1. Introduction

Soil extracellular enzyme activity (EA) is closely related to soil properties, soil types and environmental conditions, and is widely used as an important index to evaluate soil quality and soil biological activity [1]. B-1,4 glucosidase (βG), β-1, 4-N-acetylglucosidase (NAG), leucine aminopeptidase (LAP) and alkaline phosphatase (AP) can be used as indicators of soil C, N and P requirements, respectively [2,3]. These enzymes play an important role in the soil nutrient cycle and participate in a large number of soil biochemical processes. Ecological enzyme stoichiometry can predict nutrient limits of soil microorganisms to some extent and can relate extracellular enzyme activity to microbial resource allocation under

different environmental conditions [4]. From this perspective, ES can be used as a good indicator of microbial nutrient acquisition that can reflect soil environmental changes [5]. Sinsabaugh's study showed that different soil types will cause changes in EA, but the $E_{C:N:P}$ remains at 1:1:1 in different ecosystems. Deviations in the ratio indicate that it is limited by soil nutrients, such as carbon, nitrogen and phosphorus [6]. The stoichiometric imbalance ($E_{C:N:P}$) between microorganism and substrate will drive the decomposition of organic matter, that is, microorganisms change their chemical composition in response to changes in the chemical composition of the substrate through unsteady behavior to realize internal stability of the microorganism.

Soil extracellular enzyme activities involved in C, N and P turnover play a key role in soil biogeochemical processes [6]. The EA response to nitrogen fertilizer has been studied for several decades, and it generally shows changes in direction and amplitude in different studies [7]. The amount of available nitrogen in the soil is limited, the production of enzymes increases to decompose refractory organic matter, and when there is enough nitrogen, the production of enzymes will be reduced [4]. Due to the essential coupling of carbon and nitrogen cycles in terrestrial ecosystems, increased nitrogen availability can alter the formation and decomposition of soil organic matter (SOM) [8]. Nitrogen and phosphorus, as essential elements, often limit plant growth and microbial function together. Many studies have also confirmed that nitrogen application reduces the level of available phosphorus in soil [9]. Phosphorus in soil mainly exists in organic form, and alkaline phosphatase is the main enzyme involved in phosphorus cycle [10]. Based on specific stoichiometric ratios, crop growth induced by exogenous nitrogen increases plant demand for soil phosphorus, leading to a decrease in soil available phosphorus content. In order to meet the growing demand of phosphorus for plants and microorganisms, the addition of nitrogen can improve soil phosphatase activity [9].

China is a large agricultural country, and soil health is very important for sustainable development of agriculture in China. The Loess Plateau is a typical fragile eco-environment area [11]. Affected by natural and human factors, the shortage of water resources and soil fertility limit the increase and stability of agricultural production in the Loess Plateau [12,13]. Plastic film mulching can significantly reduce soil water evaporation and conserve water, which is widely used in arid and semi-arid areas [13,14]. Studies have shown that plastic film mulching will accelerate soil nutrient consumption. Compared with the relatively stable physical and chemical properties of soil, soil enzyme activity can quickly respond to the changes of soil conditions over a short time and is easily affected by the environment [15]. As one of the most active components in soil, soil enzymes play an important role in soil nutrient cycle as well as the transformation and decomposition of organic matter [16]. Therefore, it is of great significance to discuss changes in soil enzyme activities to evaluate farmland soil nutrients and soil quality.

Therefore, this study analyzed the changes in soil carbon, nitrogen and phosphorus content and the key enzyme activities of soil carbon, nitrogen and phosphorus cycle and their stoichiometric ratio; explored the soil carbon, nitrogen and phosphorus content, enzyme activities and ecological enzyme metrology characteristics as well as their internal relations; and revealed the characteristics of soil nutrients after plastic film mulching, as well as the response and regulation mechanism of soil enzyme activities to nutrient cycle. This study aimed to provide a scientific basis for the sustainability of rational fertilization under plastic film mulching measures.

## 2. Materials and Methods

This study was carried out in the Shouyang key field scientific observation experimental station of dryland agriculture, Ministry of Agriculture and Rural Affairs (37°44′52″ N, 113°12′11″ E). The average annual temperature in this area is 7.4 °C, the frost-free period is about 140 d, and the average annual precipitation in the last five years is 518 mm. Precipitation is is unevenly distributed throughout the year and occurs mainly in June, which accounts for more than 80% of the annual rainfall. The sunshine hours are 2858 h. The

area belongs to a temperate continental climate, and the experimental site is flat and has no irrigation conditions.

### 2.1. Experimental Design and Treatment

The positioning experiment began in April 2015; the sowing time occurs in May, and the harvesting time is in October. There is one sowing season each year. Field operations, such as fertilization, rotary tillage and harrowing, are completed before film mulching. A total of 150 kg·hm$^{-2}$ of superphosphate (including P$_2$0$_5$ 16%) and 75 kg·hm$^{-2}$ of potassium chloride (KCl 60%) are applied, and fertilizers are applied one at a time while sowing. The film is a white PE agricultural film with a thickness of 0.008 mm and a width of 1.2 m.

The experiment was divided into two planting modes: plastic film mulching (PM) and no plastic film mulching (LD). Five nitrogen fertilizer levels of 0, 90, 150, 225 and 300 kg·hm$^{-2}$ were set, with a total of 10 treatments, and each treatment was repeated three times, with a total of 30 communities. Each plot area was 48 m$^2$. The tested maize variety was "Jingdan 951"; row spacing was 50 cm, plant spacing was 30 cm and sowing density was 66,666 plants·hm$^{-2}$.

### 2.2. Sample Collection and Preservation

During the maize harvest in October 2021, soil samples were collected at three depths of 0–10 cm, 10–20 cm and 20–30 cm with an auger that had a diameter of 5 cm. Collection was repeated three times in each plot, and collected soil samples were left in the laboratory to remove debris, such as gravel and weeds. Part of the sample was air-dried and passed through a 2 mm sieve for the determination of soil pH; part of it was air-dried through a 100-mesh sieve to determine soil organic carbon (SOC), total nitrogen (TN) and total phosphorus (TP). Part of the sample was put into the −80 refrigerator, and the activity of soil extracellular enzymes was measured within one week.

### 2.3. Soil Chemical Analysis

The pH was determined by immersion potential method, and the water-soil ratio was 5:1. SOC was determined by potassium dichromate external heating method. A total of 0.1000 g air-dried soil, 5 mL potassium dichromate and 5 mL concentrated sulfuric acid were combined, digested at 170–180 °C and titrated with 0.2 mol·L$^{-1}$ FeSO4. TN was determined by semi-micro Kjeldahl nitrogen determination method. In total, 0.6 g air-dried soil was digested with 3 g accelerator and 8 mL concentrated sulfuric acid, and then measured using a Kjeldahl nitrogen determination instrument. TP in soil was determined by molybdenum-antimony colorimetry, 0.6 g of air-dried soil was digested with 8 mL of concentrated sulfuric acid and 10 mL of perchloric acid, and all digested liquid was transferred to a 100 mL volumetric flask for constant volume overnight. Molybdenum and antimony reagents were added to the upper liquid, and the colors were compared with a spectrophotometer.

### 2.4. Soil Extracellular Enzyme Activity and Enzyme Stoichiometry Ratio

We used the soil extracellular enzyme measurement technique reported by Tapia Torres and Steinwe to measure the activities of four kinds of soil extracellular enzymes [17,18]: β-1,4-glucosidase (βG), β-1,4-N-acetylglucosidase (NAG) and leucine aminopeptidase (LAP) and alkaline phosphatase (AP).

We weighted 2.75 g of fresh soil and added 50 mL of sodium acetate buffer (pH 5.5) to extract soil extracellular enzyme, then shook for 10 min to make a soil suspension. A total of 800 μL of soil suspension and 200 μL of fluorescently labeled C, N and P substrate solution with a concentration of 200 μmol·L$^{-1}$ were sucked into a 96-well plate and cultured in an incubator at 25 °C for 4 h in the dark. Readings were taken under the conditions of excitation wavelength of 365 nm and emission wavelength of 450 nm. The unit was nmol activity g$^{-1}$ dry soil h$^{-1}$.

Microbial C:N, C:P and N:P acquisition (EC:N, EC:P, and EN:P, respectively)

$E_{C:N} = \ln(\beta G):\ln(NAG + LAP)$
$E_{C:P} = \ln(\beta G):\ln(AP)$
$E_{N:P} = \ln(NAG + LAP):\ln(AP)$
$E_{C:N:P} = \ln(\beta G):\ln(NAG + LAP):\ln(AP)$

### 2.5. Statistical Analysis

Microsoft Excel 2019 was used to input and sort the test data, and origin 2023 software was used to draw. SPSS 19.0 was used for data variance analysis, and Canoco 5 was used for redundancy analysis (RDA).

## 3. Results

### 3.1. Soil Chemistry

The chemical properties of soil with different treatments are shown in Figure 1. During the harvest period, the contents of soil organic carbon (SOC) and total nitrogen (TN) showed a changing trend of 0–10 cm > 10–20 cm > 20–30 cm, while the contents of soil total phosphorus (TP) showed a changing trend of 20–30 cm > 10–20 cm > 0–10 cm. The contents of SOC, TN and TP in PM-treated soil were higher than those in LD-treated soil at different soil depths ($p < 0.05$). The SOC concentration in each soil depth increased as the N addition levels increased. Under the PM treatment, the SOC content in the 0–10 cm and 10–20 cm soil depths was highest when the N addition level was 225 kg·hm$^{-2}$, and the SOC content in other treatments was highest when the N addition level was 300 kg·hm$^{-2}$. The TN content in soil increased as the N addition level increased, and it was highest when the N addition level was 300 kg·hm$^{-2}$. The analysis of variance of two factors showed that N application level and plastic film mulching measures significantly affected the TN content in soil ($p < 0.05$), but there was no significant interaction between them ($p > 0.05$) (Figure 1, Table 1). With the increase of N addition levels, the TP content in soil increased continuously; it was highest when the N addition level was 225 kg·hm$^{-2}$ in 0–10 cm and 10–20 cm soil depths, and in the 20–30 cm soil depths, it was highest when the N addition level was 300 kg·hm$^{-2}$. Two-way ANOVA showed that N addition levels and plastic film mulching significantly affected soil TP content ($p < 0.05$), but there was no significant interaction between them ($p > 0.05$) (Table 1).

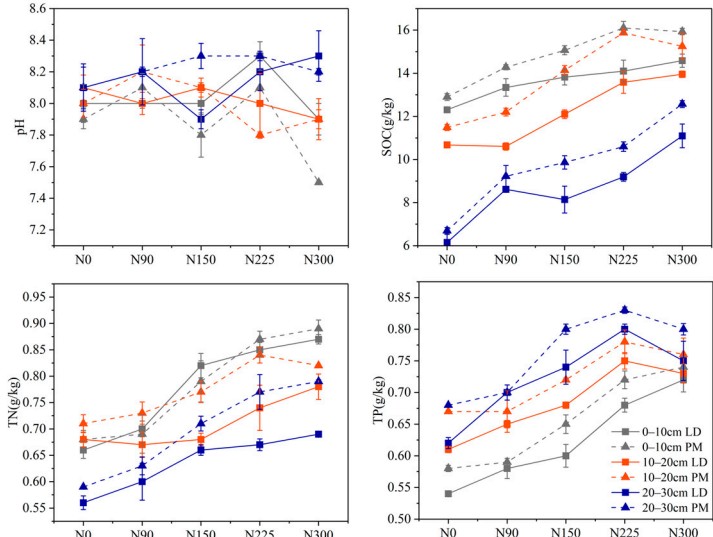

**Figure 1.** Effects of each treatment on soil physicochemical properties. SOC, soil organic carbon; TN, total nitrogen; TP, total phosphorus. N0, N addition level was 0 kg·hm$^{-2}$; N90, N addition level was 90 kg·hm$^{-2}$; N150, N addition level was 150 kg·hm$^{-2}$; N225, N addition level was 225 kg·hm$^{-2}$; N300, N addition level was 300 kg·hm$^{-2}$.

**Table 1.** Results of repeated measures ANOVA on the effects of nitrogen addition and plastic film mulching on soil properties.

| | 0–10 cm | | | | 10–20 cm | | | | 20–30 cm | | | |
|---|---|---|---|---|---|---|---|---|---|---|---|---|
| | pH | SOC | TN | TP | pH | SOC | TN | TP | pH | SOC | TN | TP |
| N | <0.01 | <0.01 | <0.05 | <0.01 | =0.09 | <0.01 | <0.01 | <0.01 | <0.05 | <0.05 | <0.05 | <0.01 |
| P | <0.01 | =0.12 | <0.01 | <0.01 | <0.01 | <0.01 | <0.01 | <0.01 | =0.17 | <0.01 | <0.01 | <0.01 |
| N*P | <0.01 | =0.13 | =0.15 | =0.43 | <0.01 | <0.01 | =0.35 | =0.33 | <0.01 | =0.94 | =0.22 | =0.08 |

SOC, soil organic carbon; TN, total nitrogen; TP, total phosphorus. N, Nitrogen addition levels; P: Plastic film mulching; N*P: Nitrogen addition level and plastic film mulching. Indicates $p < 0.01$, extremely significant different. Indicates $p < 0.05$, significantly different.

### 3.2. Soil Extracellular Enzyme Activities and Its Stoichiometry

The changes of EA under different treatments are shown in Figure 2. After 8 years of long-term location test, we found that the changes of soil enzyme activities in different soil depths were consistent. The activities of soil βG and AP increased as the N application rate increased and reached their highest values when the N addition level was 225 and 300 kg·hm$^{-2}$, respectively. At this time, the enzyme activities in soil under plastic film mulching treatment were significantly higher than those in LD soil enzyme activities ($p < 0.05$). Correlation analysis showed that plastic film mulching and different N addition levels had significant or extremely significant effects on soil enzyme activities, and there was an obvious mutual relationship between them (Table 1). The activity of NAG+LAP in soil was higher at N addition levels of 0 and 90 kg·hm$^{-2}$, and the activity of NAG+LAP in soil treated with PM was higher than that of LD. The analysis of variance of two factors showed that there was an obvious interaction between plastic film mulching and N addition levels ($p < 0.01$) (Table 2).

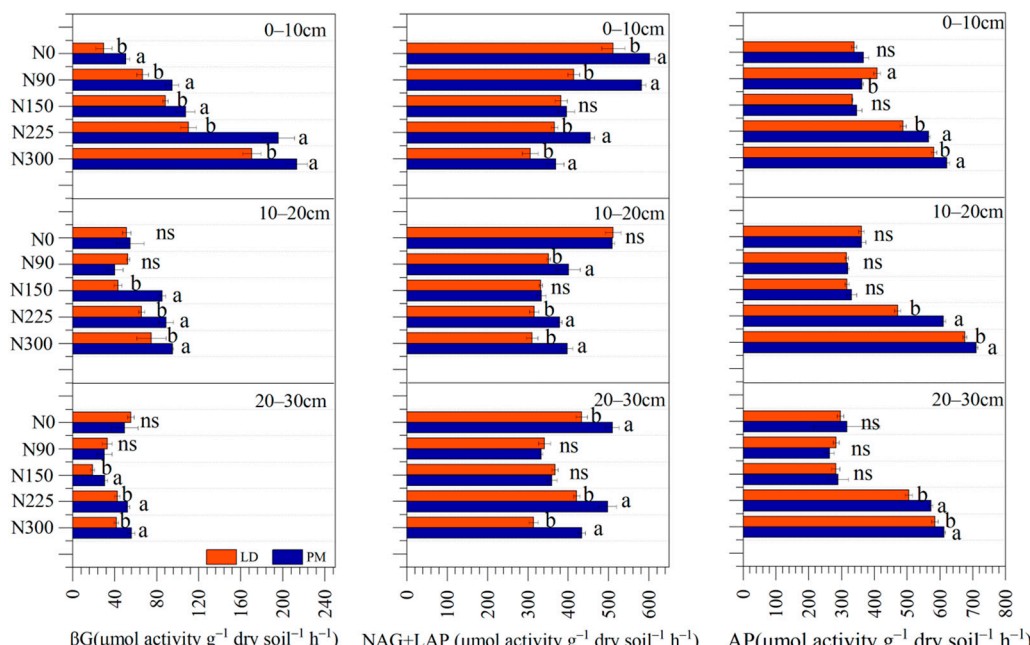

**Figure 2.** Soil enzyme activities under different treatment levels; N0, N addition level was 0 kg·hm$^{-2}$; N90, N addition level was 90 kg·hm$^{-2}$; N150, N addition level was 150 kg·hm$^{-2}$; N225, N addition level was 225 kg·hm$^{-2}$; N300, N addition level was 300 kg·hm$^{-2}$. Different lowercase letters (a, b) indicate that there was a significant difference and "ns" indicates that there was no significant difference between the PM and LD treatments.

**Table 2.** Results of repeated measures ANOVA on the effects of N addition levels and plastic film mulching on soil enzyme activities.

|  | 0–10 cm | | | 10–20 cm | | | 20–30 cm | | |
|---|---|---|---|---|---|---|---|---|---|
|  | βG | NAG + LAP | AP | βG | NAG + LAP | AP | βG | NAG + LAP | AP |
| N | <0.01 | <0.01 | <0.01 | <0.01 | <0.01 | <0.01 | <0.05 | <0.01 | <0.05 |
| P | <0.01 | <0.01 | <0.01 | <0.01 | <0.01 | <0.01 | <0.01 | <0.01 | <0.01 |
| N*P | <0.01 | <0.01 | <0.01 | <0.01 | <0.01 | <0.01 | <0.05 | <0.01 | <0.01 |

N, N application levels; P, Plastic film mulching; N*P: N application levels and plastic film mulching. Indicates $p < 0.01$, extremely significant different. Indicates $p < 0.05$, significantly different.

Under LD treatment, the values of $E_{C:N}$, $E_{C:P}$ and $E_{N:P}$ are 0.52–0.89, 0.52–0.80 and 0.84–1.08, respectively. The values of $E_{C:N}$, $E_{C:P}$ and $E_{N:P}$ under PM treatment are 0.58–0.90, 0.61–0.84 and 0.91–1.08, respectively (Figure 3 and Table 3). In this experiment, the values of soil $E_{C/N}$ and $E_{C/P}$ gradually increased with N addition, but both were less than 1 (Figure 3), indicating that soil microorganisms were limited by both N and P nutrients (Figures 3 and 4). When the N application levels are 0, 90 and 150 kg·hm$^{-2}$ and the $E_{N:P}$ is greater than 1 in general, it can be seen that soil microorganisms are mainly limited by N nutrients rather than P nutrients. When the N application levels are 225 and 300 kg·hm$^{-2}$ and $E_{N:P} < 1$, it can be seen that levels are mainly limited by P nutrient. At the same time, we found that, compared with LD, the $E_{C:N}$, $E_{C:P}$ and $E_{N:P}$ values of each soil depth under PM treatment were closer to 1:1, which indicated that plastic film mulching treatment was beneficial to soil nutrient balance.

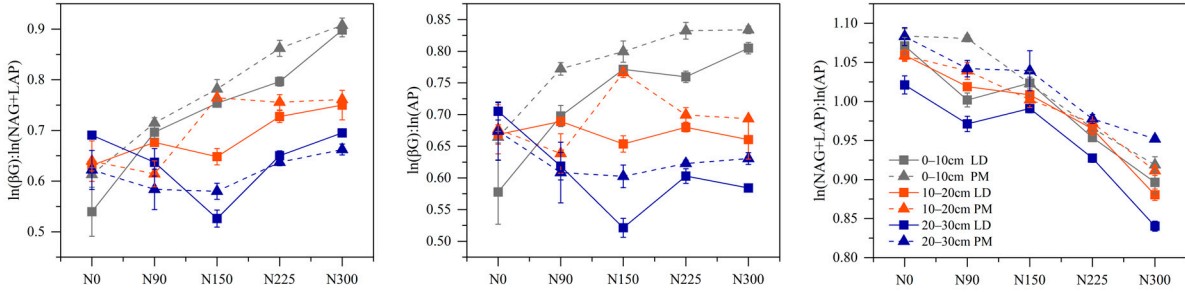

**Figure 3.** Plots of stoichiometric ratio of soil enzyme activities under different N addition levels under the treatment of PM and LD treatments.

**Table 3.** Results of repeated measures ANOVA on the effects of N addition levels and plastic film mulching on stoichiometric ratio of soil enzyme activities.

|  | 0–10 cm | | | 10–20 cm | | | 20–30 cm | | |
|---|---|---|---|---|---|---|---|---|---|
|  | $E_{C:N}$ | $E_{C:P}$ | $E_{N:P}$ | $E_{C:N}$ | $E_{C:P}$ | $E_{N:P}$ | $E_{C:N}$ | $E_{C:P}$ | $E_{N:P}$ |
| N | <0.01 | <0.01 | <0.01 | <0.05 | <0.05 | <0.05 | <0.01 | <0.05 | <0.01 |
| P | <0.01 | <0.01 | <0.01 | <0.01 | <0.01 | <0.01 | <0.01 | <0.01 | <0.01 |
| N*P | <0.05 | <0.05 | <0.01 | <0.01 | <0.01 | <0.01 | <0.01 | <0.01 | <0.01 |

N, N addition levels; P, Plastic film mulching; N*P, N addition levels and plastic film mulching. $p < 0.01$ indicates extremely significant difference, and $p < 0.05$ indicates significant difference.

In order to further clarify the demand of soil extracellular enzymes for soil nutrients under different N application levels and LD and PM measures, we conducted the following analysis (Figure 4). The results show that the soil in the experimental area is mainly limited by nitrogen and phosphorus. Specifically, as the N application levels increased, the soil gradually changed from N limitation to P nutrient limitation, and the N application levels of 150 and 225 kg·hm$^{-2}$ were more conducive to the balance of soil nitrogen and phosphorus nutrients. The $E_{C/N}$ and $E_{C/P}$ were closer to 1 when the N application levels were

225 and 300 kg·hm$^{-2}$, respectively, which indicated that this time was more beneficial to the balance of soil carbon, nitrogen and carbon, and phosphorus. Therefore, comprehensive analysis and comparison show that when the N application level is 225 kg·hm$^{-2}$, it is more conducive to the leveling of soil nutrients.

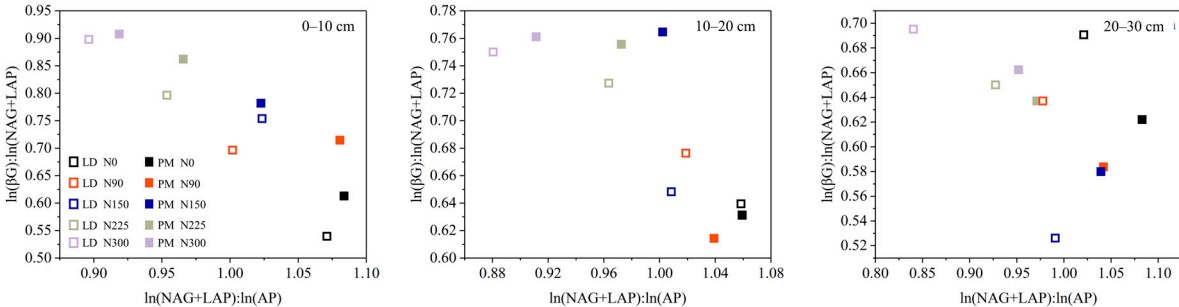

**Figure 4.** Scatter plot of ES under different N addition levels under PM and LD treatments.

### 3.3. Relationship between Soil Chemical Properties and EA and ES

In order to clarify the correlation between soil extracellular enzyme activities and soil environmental factors that participate in soil C, N and P nutrient cycling and affect the soil depths, we conducted RDA analysis on soil environmental factors (pH, SOC, TN, TP, C:N, C:P and N:P) and soil extracellular enzyme activities (βG, NAG+LAP and AP) (Figure 5). RDA analysis showed that the chemical properties of soil well explained the changes of soil enzyme activities under LD treatment (63.4% and 63.1%) (Figure 5). βG showed positive correlations with TP, TN and N:P in soil; negative correlations with SOC, C:P and C:P; NAG+LAP showed positive correlations with pH and strong negative correlations with SOC, TP and TN. AP showed significant positive correlations with organic carbon, total N and total P. $E_{C:N}$ and $E_{C:P}$ showed strong positive correlations with soil TP and N:P.

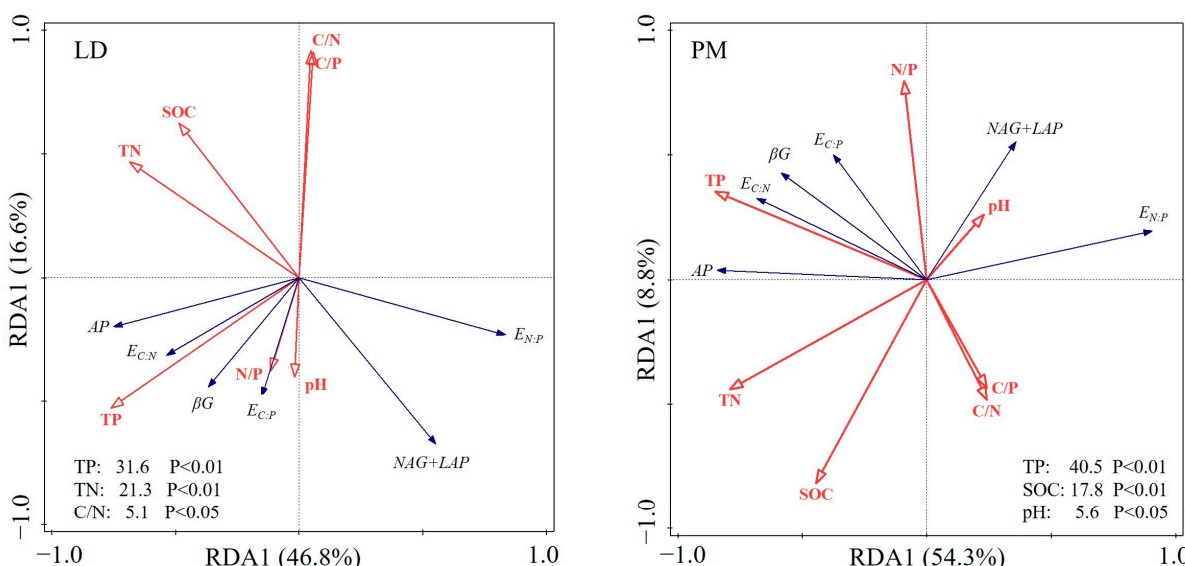

**Figure 5.** Redundancy analysis of soil chemical properties and EA and ES under different treatments.

## 4. Discussion

Soil organic carbon (SOC), total nitrogen (TN) and total phosphorus (TP) are important indicators of soil health. Our study found that the contents of SOC, TN and TP in PM treated soil were higher than those in LD treated soil ($p < 0.05$). These results show that plastic film mulching can significantly increase soil nutrient content. Ding's research results are basically consistent with our conclusions. After 28 years of nitrogen application under plastic film mulching, it was found that soil total nitrogen increased by 25% [19].

Lee et al. [20] found that plastic film mulching can significantly improve soil nitrogen mineralization and nitrogen availability. At the same time, we found that soil nutrients changed with the N addition levels. Studies have shown that the improvement of nitrogen availability can promote the decomposition of unstable organic matter in soil [21]. However, SOC has an important influence on soil physical and chemical properties such as total N, total P, available N and available K [22].

Our results showed that the soil enzyme activity under PM treatment was generally higher than that under LD treatment. At the same time, compared with LD, the $E_{C:N}$, $E_{C/P}$ and $E_{N/P}$ values of each soil depth under PM measures are closer to 1, indicating that PM is more beneficial to soil nutrient balance (Figure 2) [23]. The increase of soil enzyme activity by plastic film mulching can be analyzed from four aspects: (1) Plastic film mulching can increase the source of soil enzyme by increasing soil microbial biomass [24]. (2) A large number of studies show that soil enzyme activity and its stoichiometric ratio have strong correlation with soil nutrients. Film mulching can increase the content of C, N and P in soil [25,26]. (3) Plastic film mulching can also change environmental conditions around the soil, such as soil moisture, soil temperature and humidity, which is beneficial to the adsorption of soil enzymes and the occurrence of enzymatic reactions [13,22]. (4) The results show that plastic film mulching can effectively promote the growth of crop roots and improve the distribution of roots in soil profile. These improvements enhance the absorption of nutrients and water by maize roots from soil [27].

After 7 years of long-term location experiment, soil N addition levels had a significant effect on soil enzyme activities. We found that, during harvest, high N addition levels more strongly promoted the activities of βG and AP in soil but inhibited the activities of N-cycle related enzymes. The results showed that additional inorganic nitrogen input further increased the demand of microorganisms and crops for carbon and phosphorus nutrients. Due to the poor migration ability of soil available phosphorus, plastic film mulching accelerated the consumption of soil available phosphorus, thus increasing the activities of soil carbon and phosphorus circulating enzymes [9,28–30]. The increase of C-cycle enzyme represents the change of C-cycle rate, and the increase of C-cycle enzyme activity indicates that the decomposition speed of soil organic matter is accelerated, which may be because nitrogen application promotes the increase of litter and roots, increases the C:N ratio in soil, and increases the demand for C, thus promoting the activity of C-cycle enzyme [31,32].

At the same time, the addition of N in this experiment also led to the increase of phosphatase activity, which was also confirmed by many experts [7,29,33]. The addition of N changed the soil environment from N-limited to P-limited. Phosphatase is an enzyme rich in N, which needs to consume excess nitrogen in the process of obtaining P by forming phosphatase [15]. Therefore, the addition of nitrogen can supplement available phosphorus and fix nitrogen in soil to promote the balance of soil nutrients. This shows the self-regulation ability of soil ecosystem. In our study, the activity of NAG+LAP in soil was higher when the nitrogen application level was 0 and 90 kg·hm$^{-2}$ and decreased with the increase of nitrogen application levels. This shows that a large amount of available nitrogen in the soil can greatly slow down the limitation of soil microorganisms on nitrogen, making the production of nitrogen-related enzymes more conservative [11]. Through 8 years of nitrogen application treatment, Ajwa et al. found that nitrogen application would have a negative impact on the activity of N-circulating enzyme [34]. This demonstrates that nitrogen has a certain regulatory role in the soil ecosystem, and the shortage of nitrogen demand will stimulate the soil to produce enzymes that will ensure the acquisition of nitrogen [35].

Studies show that soil conditions directly or indirectly affect plants and microorganisms [36], so soil enzyme activities may be different due to differences in soil types or environments. Studies have also shown that, even under different soil types and climatic conditions, the stoichiometric ratio of soil ecological enzymes follows the global model, that is, $E_{C:N:P}$ is 1:1:1, When the ratio is shifted, it means that the soil is limited by C, N or P nutrients at

that time [6,17]. In our experiment, the $E_{C:N}$ and $E_{C:P}$ values of the soil are both less than 1 (Figure 3). This shows that the soil is limited by both N and P nutrients (Figures 3 and 4). When the nitrogen application levels are 0, 90 and 150 kg·hm$^{-2}$, the $E_{N:P}$ is greater than 1 in general, which demonstrates that, compared with P, the enzyme levels are mainly limited by N nutrients. When the nitrogen application levels are 225 and 300 kg·hm$^{-2}$, and $E_{N:P} < 1$, it shows that the enzyme levels are mainly limited by P nutrient. These results show that microorganisms can adjust their physiological metabolism to adapt to low N or P nutrients as well as to drought and low nutrient environments. This conclusion has been confirmed in previous studies. Nitrogen application will aggravate the limitation of phosphorus in soil [15]. Olander and Vitousek found that long-time(4–6 year) N fertilization significantly enhanced phosphatase activity [37]. Cui found that microbial nutrient metabolism is limited by both N and P nutrients in the arid and poor Loess Plateau [38]. P may be strongly bound by calcium and magnesium ions in alkaline soil, so soil phosphorus is an important limiting factor in the Loess Plateau [39]. Soil hydrothermal conditions under plastic film mulching accelerate the decomposition of soil organic carbon, and also increase the limitation of soil phosphorus [40].

The characteristics of soil chemical properties are important factors affecting soil enzyme activity and its stoichiometric ratio [18,41]. The chemical properties of soil can well explain the changes in soil extracellular enzyme activity and its stoichiometry (Figure 5). pH, SOC, TN and TP are important factors affecting the changes in soil enzyme activities, and this conclusion is consistent with previous studies [38]. In our study, βG showed positive correlation with TP, TN and N:P in soil; negative correlation with SOC, C:P and C:P; NAG+LAP showed positive correlation with pH as well as strong negative correlation with SOC, TP and TN. AP showed significant positive correlation with organic carbon, total nitrogen and total phosphorus. This result shows that soil enzyme activity is closely related to soil environmental changes, and soil chemical properties can greatly affect soil enzyme activity and its stoichiometric ratio by changing the concentration of effective substrate and soil nutrient stoichiometry, which shows that soil enzyme activity is largely controlled by soil physical and chemical properties [42]. It can also be seen that the availability of soil nutrients is the main factor affecting the nutrient ratio of microorganisms. It is worth noting that soil enzyme activity and soil nutrients generally showed a negative correlation, but in this study, soil AP enzyme activity was positively correlated with soil TP, indicating that microorganisms may secrete more acid phosphatase to hydrolyze organic phosphorus and promote the release of effective phosphorus when the effective phosphorus content is not sufficient to meet plant growth requirements [43].

## 5. Conclusions

Our results provide a direct basis for elucidating the activity and stoichiometry of extracellular enzymes in soils during harvest in arid farmlands of the Loess Plateau in northern China. Our latest findings from this study are: (1) The soil in this area is mainly limited by nitrogen and phosphorus. Considering all aspects, when the nitrogen application level is 225 kg·hm$^{-2}$, it is more conducive to the balance of soil nutrients during harvest and the promotion of soil enzyme activities. (2) Mulch increases soil enzyme activities and is more conducive to soil nutrient balance. (3) The trends of soil enzyme activities and their stoichiometry ratios in three soil depths, 0–10 cm, 10–20 cm and 20–30 cm, are basically the same. (4) The activity and stoichiometry of soil extracellular enzymes were strongly influenced by pH, SOC, TN and TP, and unlike previous studies, we found that soil AP enzyme activity was positively correlated with soil TP content in this region. In conclusion, this study reveals the self-regulatory mechanism of soil extracellular enzyme activity with N addition in arid and semi-arid regions. These findings are important for our understanding of microbial metabolism and soil nutrient cycling in the Loess Plateau ecosystem.



**Author Contributions:** Conceptualization, W.D. and M.Y.; methodology and software, M.L. and C.S.; validation, W.D. and M.W.; data curation, M.L. and M.W.; writing—original draft preparation, M.L. and H.W.; writing—review and editing, W.D.; supervision, M.Y. and X.Z.; project administration, E.L.; funding acquisition, W.D. All authors have read and agreed to the published version of the manuscript.

**Funding:** Please add: This work was supported by the National Natural Science Foundation of China (grant no. 31961143017 and 31470556), the National Key R&D Program of China (2021YFE0101300) and the Bio-Water Saving and Dry Farming Innovation Team Project of CAAS.

**Data Availability Statement:** The data provided in this study are available from the corresponding author upon request.

**Conflicts of Interest:** The authors declare no conflict of interest.

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
