# Peer review of "Self-Regulation of Soil Enzyme Activity and Stoichiometry under Nitrogen Addition and Plastic Film Mulching in the Loess Plateau Area, Northwest China"

_agriculture, doi:10.3390/agriculture13050938_

Round 1
Reviewer 1 Report
Title: suggest to indicate key findings in the title, something like "Self-regulation of soil enzyme activity and stoichiometry under nitrogen addition and plastic film mulching in the Loess Plateau area, Northwest China"
Line 16-20, this is a very long sentence, please separate it into two sentences. Also, this sentence is not clear about why setting five levels of nitrogen addition will help understand the impacts of film mulching on enzyme activity. Suggest to explain the interactions of water and nutrient retention/loss a little, and more in the Introduction.
Line 20, simply say"We found that...", and this sentence needs more work: 1. "the trend of soil enzyme activity changes" is confusion, under what levels of treatment?; 2. replace "basically the same in" with "similar among"; 3. use another sentence to discuss its influencing factors (i.e., N and P). Also, need to say among what parameters the N and P are the most limiting factors on enzyme activity.
Line 21, use "soil depth" is more appropriate than "soil layer", the latter one is more about O, A, B, C layers.
Line 25, first time introducing RDA, need to spell it out
Line 26, be cautions about the word "influence" because most statistics were just testing correlation, so the word "..were strongly correlated with..". Enzyme activity and soil total/available nutrients are influencing each other.
Line 41, delete "in recent years", how recent is this recent?
Line 46, "This study" is confusion, it is talking about literature but not this study" also add sentence explaining the meaning of enzyme stoichiometry remains constant while enzyme activities are changing.
Line 54-57, it is less meaningful to just list those findings in the past in the Introduction. what's more important is mentioning the knowledge gaps, which will emphasize the importance of this study to the community
Line 177, change the subtitle to "soil chemistry". I don't think this section contains physical properties, SOC, TN, TP, pH are all chemicals properties. Please change it throughout the manuscript.
Line 179, should be concentration but not content, content is concentration times mass
Table 1, two digits for non significant values are sufficient
Table 3, subjects in the second row are not correct, should be the ratio but not each enzyme
Line 279-285, this paragraph sounds like an Introduction section. Discussion should be centered around the findings and associated implications
Line 287, it would be interesting to standardize the enzyme activity by SOC, so really looking at traits see if there are any effects of nitrogen addition and film mulching
Line 288-289, did you analyze soil moisture and temperature? seems like these are very important parameters for this study.
Line 307, does it really change soil structure (%sand, silt, clay, soil aggregates)?
Line 378, "correlated with..."
Author Response
Dear reviewers,
We are very grateful to you for giving us an opportunity to revise our manuscript. we appreciate you very much for your positive and constructive comments and suggestions on our manuscript entitled “Self-regulation of soil enzyme activity and stoichiometry under nitrogen addition and plastic film mulching in the Loess Plat-eau area, Northwest China” (ID: agriculture-2317786)
We have studied reviewers’ comments carefully and tied our best to revise our manuscript according to the comments. The following are the responses and revisions I have made in responses to the reviewers’ questions and suggestions on an item-by-item basis. Thanks again to the hard work of the editor and reviewer!
- Line 16-20, this is a very long sentence, please separate it into two sentences. Also, this sentence is not clear about why setting five levels of nitrogen addition will help understand the impacts of film mulching on enzyme activity. Suggest to explain the interactions of water and nutrient retention/loss a little, and more in the Introduction.
Response 1: We greatly appreciate the constructive comments and suggestions. We have revised the abstract in detail and explained it in detail in the introduction. Relevant contents have been marked in detail in the text in blue.
- Line 20, simply say"We found that...", and this sentence needs more work: 1. "the trend of soil enzyme activity changes" is confusion, under what levels of treatment?; 2. replace "basically the same in" with "similar among"; 3. use another sentence to discuss its influencing factors (i.e., N and P). Also, need to say among what parameters the N and P are the most limiting factors on enzyme activity.
Response 1: Thank you for your suggestion. We will revise the contents as follows.
“The latest discovery of this study is that the activities of soil EA involved in the cycling of soil carbon (C), nitrogen (N) and phosphorus (P) are similar in different soil depth (0−10 cm, 10−20 cm and 20−30 cm). Both EC:P and EC:P in the soil in this area are less than 1: 1, indicating that the soil is limited by N and P. Comprehensive analysis showed that when the nitrogen application level was 225 kg·hm−2, it was more beneficial to the balance of soil nutrients and the improvement of soil EA at harvest. At the same time the PM measure can effectively improve the soil EA and is more conducive to the balance of soil nutrients. Redundancy analysis (RDA) analysis showed that the EA and ES were strongly correlated with pH, soil organic carbon (SOC), total nitrogen (TN) and total phosphorus (TP). Most importantly, this study revealed that the activity of extracellular enzymes in arid and semi−arid areas was constantly self−regulated with the addition of nitrogen, which provided theoretical and technical support for the efficient use of nitrogen under the condition of plastic film mulching.”
Relevant parts are marked in blue in the manuscript.
- Line 21, use "soil depth" is more appropriate than "soil layer", the latter one is more about O, A, B, C layers.
Response 3: We are very grateful for the mistake you pointed out. We have changed all "soil layer" to "soil depth" in the manuscript.
- Line 25, first time introducing RDA, need to spell it out4.
Response 4: We are very grateful for your suggestion. We have corrected it in the manuscript, and the relevant parts are marked in blue.
- Line 26, be cautions about the word "influence" because most statistics were just testing correlation, so the word "..were strongly correlated with..". Enzyme activity and soil total/available nutrients are influencing each other.
Response 5: We are very grateful for your suggestion. We have corrected it in the manuscript, and the relevant parts are marked in blue.
- Line 41, delete "in recent years", how recent is this recent?
Response 6: We are very grateful for your suggestion. We have deleted “in recent years” from the manuscript.
- Line 46, "This study" is confusion, it is talking about literature but not this study" also add sentence explaining the meaning of enzyme stoichiometry remains constant while enzyme activities are changing.
Response 7: Thank you for your suggestion. We have deleted "in recent years" and explained it. The specific changes have been marked in blue in the paper.
- Line 54-57, it is less meaningful to just list those findings in the past in the Introduction. what's more important is mentioning the knowledge gaps, which will emphasize the importance of this study to the community.
Response 8: We greatly appreciate the constructive comments and suggestions. We have deleted the original part of "54-57" and added the importance of this study to the community. The relevant contents are marked in blue in the introduction. Thank you.
- Line 177, change the subtitle to "soil chemistry". I don't think this section contains physical properties, SOC, TN, TP, pH are all chemicals properties. Please change it throughout the manuscript.
Response 9: Thank you very much for your suggestion. We have revised the subtitle 3.1 of the paper.
- Line 179, should be concentration but not content, content is concentration times mass
Response 10: Thank you very much for pointing out the problem, and we have revised it in the paper.
- Table 1, two digits for non significant values are sufficient1.
Response 11: We greatly appreciate the constructive comments and suggestions, which has been revised in detail in Table 2. Relevant parts are marked in blue.
- Table 3, subjects in the second row are not correct, should be the ratio but not each enzyme.
Response 12: Thank you very much for pointing out the error. We have revised the error in Table 3. Relevant parts are marked in blue. thank you.
- Line 279-285, this paragraph sounds like an Introduction section. Discussion should be centered around the findings and associated implications.
Response 13: We are very grateful for your comments. We have deleted relevant parts from the manuscript.
- Line 287, it would be interesting to standardize the enzyme activity by SOC, so really looking at traits see if there are any effects of nitrogen addition and film mulching14.
Response 14: We greatly appreciate the constructive comments and suggestions. so we can make relevant changes, such as the content is marked in blue in the manuscript.
- Line 288-289, did you analyze soil moisture and temperature? seems like these are very important parameters for this study.
Response 15: I'm very sorry. In our study, we focused on exploring the difference of soil nutrients caused by plastic film mulching and no plastic film mulching, and made clear the change law of soil EA and ES at different nitrogen addition levels after plastic film mulching. It has been proved by many experts and scholars that film mulching can improve soil water content and surface soil temperature in early spring. Therefore, we didn't measure and analyze the soil humidity and temperature in the experiment. In the future, we will collect and analyze relevant data. Thank you.
- Line 307, does it really change soil structure (%sand, silt, clay, soil aggregates)?
Response 16: According to your query, we consulted the relevant literature and found that the soil structure could not be changed in a short time, so we have deleted the “soil structure” in the manuscript. I am very sorry.
- Line 378, "correlated with..."
Response 17: We greatly appreciate the constructive comments and suggestions. so we can make relevant changes, such as the content is marked in blue in the manuscript.

Reviewer 2 Report
The obtained results were well developed, both in terms of content and graphics. The disadvantage of the work is only one-time? collection of soil samples for analysis.
Comments:
· There is no precise information about the years of research, such information should be included in the abstract and in the methodology. The information provided is inaccurate:
- line 88 – This study was conducted in 2015
- line 108 - taking samples for testing: October 2021? or just a one-time test
- Line 313 -After 8 years of long-term location experiment - there was no information before (in the abstract and methodology) - how the 8-year period was calculated, since there are only 6 years between 2015-2021
· How was corn grown?; in monoculture or in rotation after other plants
· Was the one-time sampling in 2021 sufficient and thus whether the tests are representative? The activity of enzymes in the soil is subject to large fluctuations during the growing season of plants.
· How to explain a fairly large increase in the content of total C and P and organic C under the influence of higher doses of N fertilization (up to 300 kg) - is this confirmed in the literature?.
· Lack of explanation of abbreviations under figures and tables - it makes interpretation difficult
· How is only pH can you write about physical properties (Figure 5)
· No DOI numbers in the reference list
· Lack of uniform spelling of the titles of cited works in references (upper or lower case)
Author Response
Dear reviewers,
Thank you very much for giving us the opportunity to revise the manuscript. We appreciate your constructive comments on the manuscript of "Self-regulation of soil enzyme activity and stoichiometry under nitrogen addition and plastic film mulching in the Loess Plat-eau area, Northwest China"(ID: agriculture-2317786 ), and thank you for your affirmation of our work on this manuscript, which makes us very happy.
We have studied reviewers’ comments carefully and tied our best to revise our manuscript according to the comments. The following are the responses and revisions I have made in responses to the reviewers’ questions and suggestions on an item-by-item basis. Thanks again to the hard work of the editor and reviewer!
The obtained results were well developed, both in terms of content and graphics. The disadvantage of the work is only one-time? collection of soil samples for analysis.
Comments:
- There is no precise information about the years of research, such information should be included in the abstract and in the methodology. The information provided is inaccurate:
line 88 – This study was conducted in 2015
line 108 - taking samples for testing: October 2021? or just a one-time test
Line 313 -After 8 years of long-term location experiment - there was no information before (in the abstract and methodology) - how the 8-year period was calculated, since there are only 6 years between 2015-2021
Response 1: Thank you for your questions. In view of the above problems, we have carefully supplemented them in the abstract and methods. In this study, planting began in April, 2015, and a long-term plastic film mulching fertilization experiment was conducted. By October, 2021, the harvesting period of spring corn had been planted for seven years (2015, 2016, 2017, 2018, 2019, 2020, 2021). We have marked the revised parts in gray in the manuscript. Thank you very much for your advice.
- How was corn grown?; in monoculture or in rotation after other plants。
Response 2: We use one crop a year, and start planting at the end of April and harvest in October every year. This content is supplemented in the manuscript. See 2.1 Experimental Design and Treatment for details.
- Was the one-time sampling in 2021 sufficient and thus whether the tests are representative? The activity of enzymes in the soil is subject to large fluctuations during the growing season of plants.
Response 3: In response to the expert's question that the soil enzyme activity fluctuates greatly at different growth stages during the plant growing season, our explanation is as follows: The first experiment started in 2015, and each treatment was repeated three times. Up to now, the field planting experiment has been carried out for 7 years, and the soil environmental conditions and microbial conditions are basically stable. At the same time, we sampled three times in different soil layers (0−10cm, 10−20cm and 20−30cm) during the corn harvest in 2021, which is enough to explain the response mechanism of soil enzyme activities to plastic film mulching and nitrogen application level. Moreover, we also analyzed the changes of soil extracellular enzyme activities in different growth stages of maize, and the corresponding analysis has been published.
After that, we will adopt your opinion and conduct long-term analysis by sampling for many years. Thank you for your advice.
- How to explain a fairly large increase in the content of total C and P and organic C under the influence of higher doses of N fertilization (up to 300 kg) - is this confirmed in the literature?.
Response 3:In our study, with the increase of nitrogen application rate, the total nitrogen content of soil has been increasing. When the nitrogen application level was 300 kg·hm−2, the total phosphorus content in 10−20 cm and 20−30 cm soil was lower than that in 225 kg·hm−2 soil. This is because high nitrogen promotes the demand of crops and soil microorganisms for soil phosphorus, resulting in the decrease of phosphorus content in soil[1].
At the same time, when the nitrogen application level was 300 kg·hm−2, the SOC content in soil did not keep increasing all the time. The SOC content in 0−10 cm and 10−20 cm soil layers decreased under PM treatment, while the SOC content in 0−10 cm and 10−20 cm soil depth under LD treatment and 20−30 cm soil under LD and PM treatment gradually increased with the nitrogen application rate.
It was found that a higher level of nitrogen application could promote the decomposition of organic matter by reducing C:N in soil, and plastic film mulching was more conducive to creating a suitable microenvironment and promoting microbial metabolic activity, so the content of organic matter in soil decreased[2]. At the same time, the decomposition of soil organic carbon has an important impact on soil nutrients, increasing the content of soil nutrients such as total nitrogen, total phosphorus and available nitrogen[4]. This may be the reason for the decrease of SOC content in 0−10 cm and 10−20 cm soil layers under PM treatment.
At the same time, nitrogen application will also increase the input of soil organic carbon from aboveground parts, which compensates for the decrease of underground carbon distribution caused by nitrogen application, thus increasing the organic carbon content in soil. Some studies believe that the higher microbial biomass stimulated by fresh litter is closely related to the formation of soil humus, and may eventually be preserved in the soil for a long time[4]. Therefore, nitrogen application will be more conducive to the formation of soil humus and promote the increase of soil carbon pool. This may be the reason why the soil depth of 0−10 cm and 10−20 cm under LD treatment and the SOC content of 20−30 cm under LD and PM treatment gradually increased with the nitrogen application rate.
- Hou, E.; Chen, C.; Wen, D.; Liu, X., Phosphatase activity in relation to key litter and soil properties in mature subtropical forests in China. Science of the Total Environment 2015, 515−516, 83−91. DOI: 10.1016/j.scitotenv.2015.02.044
- Donald, R. Z.; William, E. H.; Andrew, J. B.; Kurt, S. P.; Alan, F. T. Simulated atmospheric NO3− deposition increases soil organic matter by slowing decomposition. Ecological Applications 2008, 18(8), 2016–2027. DOI: 10.1890/07−1743.1
- Li, Y.; Wu, L.; Zhao, L.; Lu, X.; Fan, Q.; Zhang, F. Influence of continuous plastic film mulching on yield, water use efficiency and soil properties of rice fields under non−flooding condition. Soil and Tillage Research 2007, 93 (2), 370−378. DOI: 10.1016/j.still.2006.05.010
- Kirkby, C.A.; Richardson, A.E.; Len J.W.; John B. Passioura a, Graeme D. Batten b, Chris Blanchard b, John A. Kirkegaard a.Nutrient availability limits carbon sequestration in arable soils. Soil Biology and Biochemistry 2014, 68: p. 402−409. DOI: 10.1016/j.soilbio.2013.09.032
- Lack of explanation of abbreviations under figures and tables − it makes interpretation difficult
Response 5: We greatly appreciate the constructive comments and suggestions. we which have been revised in detail in the Introduction section of the manuscript. The modified parts have been marked in red font.
- How is only pH can you write about physical properties (Figure 5)
Response 6: Thank you for pointing out the problems. According to another expert's opinion, we have changed all the expressions of "the influence of physical and chemical properties on soil extracellular enzymes and their stoichiometric ratio" to "the influence of physical and chemical properties on soil extracellular enzymes and their stoichiometric ratio".
- No DOI numbers in the reference list
Lack of uniform spelling of the titles of cited works in references (upper or lower case)
Response 7: We are very grateful for constructive comments and suggestions. We have revised the relevant parts of the reference in detail. Thank you for your valuable comments on our manuscript.

Round 2
Reviewer 2 Report
The comments and suggestions I made in the review have been fully explained and included in the revised version of the manuscript.